# Associations between Mindfulness, Executive Function, Social-Emotional Skills, and Quality of Life among Hispanic Children

**DOI:** 10.3390/ijerph17217796

**Published:** 2020-10-24

**Authors:** Chien-Chung Huang, Shuang Lu, Juan Rios, Yafan Chen, Marci Stringham, Shannon Cheung

**Affiliations:** 1School of Social Work, Rutgers University, 390 George St, New Brunswick, NJ 08901, USA; huangc@ssw.rutgers.edu (C.-C.H.); yafan.chen@rutgers.edu (Y.C.); ms2728@scarletmail.rutgers.edu (M.S.); sc1552@scarletmail.rutgers.edu (S.C.); 2Department of Social Work and Social Administration, The University of Hong Kong, Pokfulam, Hong Kong; 3Department of Sociology, Anthropology and Social Work, Seton Hall University, South Orange, NJ 07079, USA; juan.rios@shu.edu

**Keywords:** Hispanic children, mindfulness, quality of life, executive function, social-emotional skills

## Abstract

Hispanic children constitute the largest ethnic minority in the United States of America, and yet few studies examine the relationship between mindfulness and Hispanic children’s quality of life. This 2018 study seeks to gain insight into how mindfulness is associated with Hispanic children’s quality of life. We surveyed 96 children in 5th- and 6th-grade classes in three Northern New Jersey elementary schools in 2018. Structure Equation Modeling was used to examine the associations between mindfulness, executive function, social-emotional skills, and quality of life. The results indicate that mindfulness is significantly and directly associated with executive function (β = 0.53), and that executive function is positively associated with social-emotional skills (β = 0.54) and quality of life (β = 0.51) of the sampled Hispanic children. The total effects on quality of life are significant for mindfulness (β = 0.33), executive function (β = 0.62), and social-emotional skills (β = 0.20). The findings shed light upon factors that can affect Hispanic children’s quality of life and call for interventions related to these factors in order to improve their well-being.

## 1. Introduction

Hispanic American is the largest ethnic/racial minority in the United States. As of 1 July 2018, there were 59.9 million people (18.3% of the nation’s total population) who identified as Hispanic in the United States [1]. Hispanic children account for an even larger proportion of the population of children in the U.S. with 24.9% of the children under 18 claiming Hispanic origins [2].

Although there is considerable diversity in the definition of “Hispanic” or “Latino/a”, with different populations experiencing differing rates of depression, socioeconomic statuses, and cultural elements [3], many of the challenges are similar for children across the various Hispanic communities. Hispanic children in the United States often face stressors that other youth in the country do not: attending high-poverty schools [4,5], confronting discrimination [6], dealing with generational acculturation gaps and family obligations [7,8,9], contending with a lack of access to health care and struggling with language barriers [8]. Each of the above can affect quality of life and school performance. Children who experience such stressors may also struggle with self-regulatory skills, exhibit externalizing behaviors, deal with mental health issues, and underperform academically [10,11,12].

Mindfulness, the ability to intentionally observe the present moment without reacting or passing judgment, has been shown to decrease levels of stress and anxiety, increase school-based competence, and enhance well-being [10,13,14,15]. Executive Function (EF) refers to the inter-related psychological processes that are responsible for goal-directed behavior [16]. These processes control organization, planning, and emotional regulation. During childhood and adolescence, executive functioning, mediated by prefrontal cortical development, undergoes rapid growth. This growth significantly impacts a child’s social-emotional development [17,18], which determines a child’s ability to handle interpersonal situations, have empathy for others, make responsible decisions, maintain positive relationships, set and achieve goals, and understand and manage emotions [19]. The growth of these skills is essential for healthy child development and positive life outcomes [20,21,22].

Quality of life (QoL) refers to the standard of living, including the facets of educational, health and well-being, and financial development [23,24]. Quality of life involves an individual’s perception of their position within their respective culture and value system, as well as an individual’s expectations for their goals, standards, and concerns [25,26]. Increasingly, QoL is becoming recognized in psychological and medical literature as a key indicator of an individual’s psychological and physical well-being [24,26]. A subcategory of quality of life, health-related quality of life (HRQoL), focuses on self-rated health in the domains of physical health, psychological health, social health, and role performance [27]. More and more research has turned its focus on HRQoL among children and adolescents since they are engaged in a rapid process of development. In considering the associations among physical, psychological, and emotional health, scholars use HRQoL as a comprehensive health index to evaluate the quality of life of children and adolescents [25,28].

As HRQoL attracts more research attention, effective interventions to increase children’s HRQoL have also become a pressing research topic. Mindfulness-based interventions have been used to reduce stress and improve mental health [29], two facets of HRQoL [30]. The effectiveness of mindfulness to improve HRQoL has been tested with bilingual populations. Roth and Robbins [31] found that an 8-week MBSR intervention significantly improved the HRQoL of 68 participants (48 Spanish-speaking and 20 English-speaking). Although there are a small number of studies about HRQoL in ethnic minority groups, it remains an under-explored subject for Hispanic children. 

Given above literature, it is likely that mindfulness increases level of self-awareness and executive function of students [14,32], which are vital for their social-emotional skills and development [33]. Social-emotional skills are also crucial for positive person-environment interactions and academic performance, as well as reducing behavioral and emotional problems [20,22]. The improved social-emotional skills are then likely to increase quality of life of students [23,34]. In this study, we were interested in investigating how mindfulness affects quality of life. Quality of life is closely linked to individuals’ overall living standards, and an important goal for child development is achieving a high quality of life [23,24]. The hypothesized model, as illustrated in Figure 1, shows a path diagram where we hypothesized that mindfulness, in conjunction with control variables (e.g., student characteristics such as age, immigrant generation, and gender), would be positively associated with executive functioning. Executive functioning and control variables, in turn, would positively relate to social-emotional skills. Social-emotional skills and control variables would be positively associated with quality of life. Figure 1 shows a full mediation model in a way that both executive function and social-emotional skills mediate the association between mindfulness and quality of life. Alternatively, the null hypothesis is that there is no relationship between mindfulness, executive function, social-emotional skills, and quality of life. We tested our hypothesis by studying a sample of Hispanic children, with the aim of informing potential interventions that will increase the quality of life of this population.

## 2. Materials and Methods

### 2.1. Data

Through local board of education referrals and availability sampling, four 5th grade classes, and three 6th grade classes in three elementary schools from a school district with large Latino communities in northern New Jersey were selected for this study. Three of the seven classes were bilingual (teaching in Spanish and English), and the remaining four were taught solely in English. A 30-min group-administered survey was conducted in classroom settings in January of 2018. The total sample size of these seven classes was 122 students, 100 of whom self-identified as Hispanic (82% of the sample), and the other 22 children were White, Black, Asian, or other racial groups. As the target of this study is on Hispanic children, we focused on the 100 self-identified Hispanic children. Four of them had missing information for key variables and were excluded from the analysis, leading to a final sample size of 96 Hispanic students. In order to account for demographic differences between the 96 Hispanic students and the rest of the students, we ran our analysis on the full 122 student sample. The analysis showed no significant differences in demographic characteristics, such as gender and age, and key outcome variables, including mindfulness, executive function, social-emotional skills, and quality of life, between the original sample and the final sample. All procedures were approved by the Rutgers University institutional review board. English- and Spanish-version of parental and child informed consent were distributed to and obtained from all participants.

### 2.2. Measures

The dependent variable in this study is quality of life, measured by the PedsQL Generic Core Scale, a self-reported scale of 23 items that was designed to measure the four dimensions of child development delineated by the World Health Organization (WHO): physical functioning (eight items), emotional functioning (five items), social functioning (five items) and school functioning (five items) [35]. In the scale, children would rate the frequency from 0 (“never) to 4 (“almost always”) in the past month that they had experiences such as: “I worry about what will happen to me,” “I feel sad or blue,” “It is hard to pay attention in class,” “It is hard for me to walk more than one block,” and “I have trouble getting along with other kids.” We used the PedsQL Generic Core Scale to measure our dependent variable–quality of life. After reverse coding, the scale ranged from 0 to 92 where higher scores indicated higher levels of quality of life. The Cronbach’s alpha of the scale reached 0.84 in this study, which is comparable to a previous validation study among a sample of 960 children ages 5–18 years (Cronbach’s alpha = 0.88) [36].

Mindfulness was our main independent variable and was measured by the Child and Adolescent Mindfulness Measure (CAMM) [37,38]. This ten-item scale measures non-judgmental and non-avoidant responses to feelings and thoughts as well as present moment awareness [37,38]. On the CAMM scale students report the daily frequency, from 0 (“never true”) to 4 (“always true”) for: “I stop myself from having feelings that I don’t like,” “I think about things that have happened in the past instead of thinking about things that are happening right now,” “I push away thoughts that I don’t like,” “At school, I walk from class to class without noticing what I’m doing,” “I get upset with myself for having feelings that don’t make sense,” “I keep myself busy so I don’t notice my thoughts or feelings,” “It’s hard for me to pay attention to only one thing at a time,” “I think that some of my feelings are bad and that I shouldn’t have them,” “I tell myself that I shouldn’t feel the way I’m feeling,” and “I get upset with myself for having certain thoughts.” For our analysis we recoded the scale to 4 being “never true” and 0 being “always true”. The sum of the scores could range from 0 to 40 with the higher scores representing a higher level of mindfulness. The Cronbach’s alpha of the scale was 0.76 in this study. This is slightly lower compared with a previous validation study of the 10-item CAMM among non-clinical adolescents (Cronbach’s alpha = 0.84) [39].

A 36-item, self-reported scale named the Delis Rating of Executive Function (D-REF) measured Executive functioning in our study. The D-REF examines three dimensions of executive functioning: emotional functioning, behavioral functioning, and executive functioning, for children from 5 to 18 years old [40]. Examples of the items include: “I get confused when I have two or more things to do at the same time” (executive functioning), “I try to control my anger, but I just can’t” (emotional functioning), and “I wish I would think more before acting” (behavioral functioning). On the original scale students would self-report the frequency of various experiences from 1 (“seldom/never”) to 4 (“daily”). After reverse coding, the sum scores could range from 36 to 144 where higher scores represented higher levels of executive functioning. The Cronbach’s alpha of the scale reached 0.93 in this study, which falls in the range of a previous D-REF validation study (Cronbach’s alpha ranges 0.91–0.99 for self-, teacher-, and parent-report forms) [41].

In order to measure social-emotional skills we used the Social and Emotional Skills scale; a self-report measure developed by Child Trends for elementary school children [42]. The 14-item scale measures social-emotional skills in four major areas: academic self-efficacy (an individual’s belief that they can achieve intended learning outcomes), persistence (an individual’s ability to continue working on a task despite challenges), self-control (an individual’s ability to manage their behaviors and emotions while inhibiting negative responses), and mastery orientation (an individual’s self-initiated determination to enhance their competencies and skills over time). Although Child Trends did not report Cronbach’s alpha for the whole scale, it reported Cronbach’s alpha between 0.65 and 0.83 for the four areas. The Cronbach’s alpha was 0.83 in this study. The items included statements such as: “I do my school work because I enjoy it,” “I calm down quickly when I get upset,” “I always work hard to complete my school work,” and “I can wait in line patiently” [42]. Children reported how frequently they experienced these items during the past month where 1 was “*not at all like me*” and 4 was “*a lot like me*”. The sum of the items on the scale ranged from 14 to 56 where the higher scores indicate better social-emotional skills.

Information about immigrant generation, participants gender, and age was gathered to control for demographic variation. Immigrant generation was coded based on the parents’ places of birth as a categorical variable. If the child’s parents were born in the U.S., the child was coded as a 3rd-generation immigrant. They were coded as a 2nd-generation immigrant if the child was born in the U.S. but their parents were not. The 1st generation coding was used if neither the parents, nor the child were born in the U.S.

### 2.3. Analytical Strategy

Using descriptive analysis, we displayed sample characteristics such as age, immigrant generation, and gender. We used Pearson correlation analysis to determine the association between variables. We then used Structural Equation Model (SEM) to examine the hypothesized model. Figure 1 shows a full mediation model in a way that both executive function and social-emotional skills mediate the association between mindfulness and quality of life. Indices used to evaluate the model fit included: the chi-square test (*p* < 0.05 indicates significance), the comparative fit index (CFI where a value that exceeds 0.90 means goodness of fit), and the standardized root mean square residual (SRMR, where the value ideally should be less than 0.07). The Skewness/Kurtosis tests were performed to check normality of outcome variable, quality of life, and the test showed that the null hypothesis of normality cannot be rejected (*p* > 0.05).

## 3. Results

### 3.1. Descriptive Results

The descriptive statistics for our study sample is shown in Table 1. Out of our 96-student sample, 46.9% were female, 53.1% were male, and the average age was 11.2. The sample was divided into 1st-generation immigrants (52.1%), 2nd-generation immigrants (36.5%), and 3rd-generation immigrants (11.5%). With a range of 0 to 40, the mean mindfulness score was 22.7. With scores ranging from 36 to 144, the mean executive function score was 106.9. On a scale of 14 to 56, the students demonstrated moderate levels of social-emotional skills with *M* = 42.5 and *SD* = 7.7. In the range of 0 to 92, the mean quality of life score was 69.2.

### 3.2. Correlation Analysis

The Pearson correlation analysis of variables, with Bonferroni adjustment of multiple comparison, can be found in Table 2. There was a significant correlation between quality of life and executive function (r = 0.64, *p* < 0.001), social emotional skills (r = 0.45, *p* < 0.001), and mindfulness (r = 0.39, *p* < 0.01). Executive function was significantly correlated with both social-emotional skills (r = 0.52, *p* < 0.001) and mindfulness (r = 0.54, *p* < 0.001). First-generation immigrant children were associated with age (r = 0.33), demonstrating that in this sample older students were more likely to be first-generation immigrants. As shown in Figure 1, the findings of the correlation analyses were consistent with and supportive of the hypothesized model. 

### 3.3. Structural Equation Modeling Results

The chi-square results of Figure 1 based on SEM indicated that X^2^ = 34.1 (*p* < 0.001) and suggest the hypothesized model did not fit well with the data. Further analysis showed that there was a direct link between executive function and quality of life. The link was added into the model and the revised model fit well with the data as indicated by the model fit indices. SRMR was 0.02, CFI was 0.983, and chi-square results were not significant (X^2^ = 4.1, *p* = 0.13). The full decomposition of standardized direct and indirect effects of the revised model is shown in Table 3. For simplicity, we draw the revised model, without control variables, in Figure 2. The results in Table 3 appear to confirm that a direct relationship exists between mindfulness and executive function, as hypothesized, indicating a positive and direct association between mindfulness and executive function (β = 0.53, *p* < 0.001). Similarly, the reported estimates also demonstrate a direct and positive association between executive function and social-emotional skills (β = 0.54, *p* < 0.001). The results showed direct associations between executive function and quality of life (β = 0.51, *p* < 0.001) and between social-emotional skills and quality of life (β = 0.20, *p* < 0.001). The sample did not demonstrate a direct association between executive function or social-emotional skills and age, gender or generation. However, we did find that, as compared to third-generation students, being a first- or second-generational immigrant had both a direct and negative association with quality of life (β = −0.25, *p* < 0.05 and β = −0.37, *p* < 0.01, respectively).

Findings from the sample are also supportive of our hypothesized indirect and positive association between mindfulness and social-emotional skills (β = 0.29, *p* < 0.001) and mindfulness and quality of life (β = 0.33, *p* < 0.001). In addition, the results show that beyond the direct association with quality of life, executive function was also indirectly and positively associated with quality of life (β = 0.11, *p* < 0.05) through its positive association with social-emotional skills. Age, gender, and immigrant generational status did not show significant indirect effects on either social-emotional skills or quality of life.

In sum, the SEM results for this sample demonstrate that executive function was strongly and positively associated, through both direct and indirect effects, with both social-emotional skills and quality of life. The results also show that mindfulness, through its strong association with executive function, was also positively associated with social-emotional skills and quality of life. The total effects on quality of life were significant in association with mindfulness (β = 0.33, *p* < 0.001), executive function (β = 0.62, *p* < 0.001), and social-emotional skills (β = 0.20, *p* < 0.05). The reported effects of mindfulness and executive function on quality of life were large and significant, underscoring the potential positive impact of mindfulness and its concordant effect on executive function as a pathway to positively link to both social-emotional skills and quality of life for Hispanic children.

## 4. Discussion 

Hispanic students begin school with similar capabilities and cognitive processes as their peers of other ethnicities [43]. Despite this, Hispanic youth have the lowest high school and college graduation rates when compared to Asian, Black, and White students [5]. Research indicates that there is not one particular developmental problem facing Hispanic youth, but rather their lower school achievement stems from the cumulative effect of a number of environmental risk factors [44]. Hispanic children and adolescents in the U.S. face stressors because of their minority status, particularly for those born outside the U.S. These stressors include increased poverty rates, lack of health care, challenges during the migration and acculturation process, discrimination, and cultural and language barriers [6,9,45].

The aforementioned difficulties faced by Hispanic youth, especially immigrant youth, can lead to emotional symptoms [46,47,48]. A study of 70 first- and second- generation Hispanic immigrant children showed that nearly 25% of the sampled adolescents expressed serious depression and suicidal ideation [49]. The main contributing factors were family dysfunction, acculturative stress, and uncertainty about the future. A more recent study also found higher incidence of depression among Hispanic children than among other races and ethnicities in the U.S. [4]. These challenges require cultural sensitivity and ingenuity in order to help Hispanic children achieve the best quality of life possible.

Additionally, Hispanic youth utilization of mental health services was only 11.6% compared to 23.9% of European American Youth [50]. Many Hispanic parents never seek clinical help for their children’s emotional disorders because of mental health stigma, underemployment and lack of access to adequate healthcare, or fears of law enforcement and/or deportation [50]. Our results suggest that in order to support Hispanic youth who are at heightened risk of school problems and life stressors, mindfulness trainings, which can be provided in school environments, may improve youth executive function, social-emotional problems, and overall quality of life.

Self-regulation, linked with social competence and academic achievement, involves modulating behaviors, thoughts, and feelings [11]. One of the key elements of self-regulation is executive function. Executive function refers to many related, but distinct, cognitive processes including working memory, cognitive flexibility, and inhibitory control [51,52,53]. Executive function is critical in supporting children’s social-emotional adjustment and cognitive learning, which in turn influences their behaviors and academic pursuits [54]. Likewise, social-emotional skills are central to a child’s personal, social, and academic life, and these skills influence a child’s overall intelligence [34].

Although most people experience daily awareness of their surroundings, mindfulness is an enhanced attention to one’s current reality [55]. If an individual is ruminating or is stuck in the past, is anxious about the future, or their attention is divided by focusing on multiple tasks, mindfulness can help the individual to disengage from these unhealthy behavioral patterns and automatic thoughts [55]. By non-judgmentally observing their internal and external experiences, individuals can improve their physical and psychological well-being [29]. Regular mindfulness training and practice can improve sustained emotional regulation, attention, and social skills, as well as reduce stress [11,15,32]. Children practicing mindfulness are able to self-soothe and become more present. Mindfulness training can also improve control over impulses, increase self-awareness, and decrease emotional reactivity [12]. Thus, it is likely that mindfulness has positive effects on executive function and social-emotional skills, and then increase overall quality of life [32,56,57].

The findings from this study show a direct association between mindfulness and executive function and that executive function has a positive association with social-emotional skills among the sampled Hispanic children. Both executive function and social-emotional skills are positively associated with quality of life. This is in line with previous research that showed an association between teaching mindfulness and a subsequent increase in participants’ quality of life. This study also showed that immigrant generation directly affected Hispanic youth’s quality of life. First- and second-generation immigrant students had a lower quality of life than third-generation immigrants. Thus, the length of time living in the U.S. and the number of family generations born in the U.S. should be taken into account when developing interventions to improve quality of life for Hispanic children.

This study’s findings on the links between immigration generation status, executive functioning, social skills, mindfulness, and quality of life among Hispanic students fill a need in the literature to better understand strengths-based constructs, such as mindfulness, among minority Hispanic populations. While the United States has an increasing number of ethnic minority populations, educators for at-risk populations need strengths-based interventions that can be used to target their students’ specific needs. Instead of focusing on pathology, approaches such as mindfulness will allow educators to focus on and build students’ strengths [12].

Although this study establishes a connection between mindfulness and an increase in quality of life for Hispanic youth, there were limitations with our approach. For example, our data collection relied on convenience sampling. Future studies could be expanded to include a more representative sample with high statistical power. A larger sample could also further test the effects of gender, age, and immigration generation, although we did not find significant correlations between these characteristics and social-emotional skills and executive function. In addition, data gathered on key variables such as mindfulness, executive function, social-emotional skills, and quality of life were all from participant self-report. Self-reporting leaves our data subject to unintended and intended reporting errors, including social desirability bias. Future studies might consider triangulating findings from different data sources, such as using biological measurement of executive function and teacher-reported social-emotional skills of the students. Finally, a causal relationship between mindfulness, executive function, social-emotional skills, and quality of life could be established by longitudinal data and experimental design.

## 5. Conclusions

This study provides evidence that mindfulness and executive function have strong associations with social-emotional skills and quality of life among Hispanic children. This finding calls for further study of the effects of mindfulness-based practice with Hispanic children and, subsequently, their executive function and life outcomes. Such practice may incorporate mindfulness training and target high-risk groups such as the first- and second-generation immigrants, to increase child executive functioning. Key components of practice may include non-judgmental internal and external awareness and mindfulness skills training [32,58].

## Figures and Tables

**Figure 1 ijerph-17-07796-f001:**
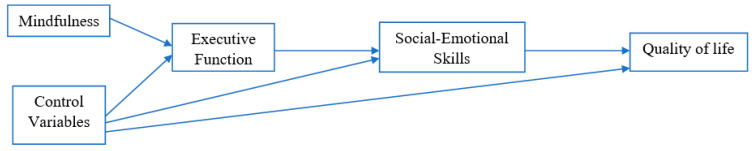
Hypothesized SEM Model.

**Figure 2 ijerph-17-07796-f002:**
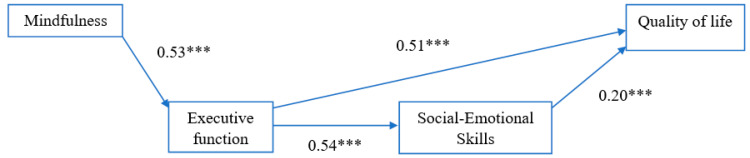
Standardized Direct Effects of Main Variables in SEM; Note: X^2^ = 4.1, *p* = 0.13. *** *p* < 0.001.

**Table 1 ijerph-17-07796-t001:** Descriptive statistics of sample characteristics.

N = 96	
Age (mean ± SD)	11.2±1.1
Gender	
Male	53.1%
Female	46.9%
Immigrant Generation	
First	52.1%
Second	36.5%
Third	11.5%
Mindfulness (mean ± SD)	22.7 ± 8.0
Executive Function (mean ± SD)	106.9 ± 12.4
Social-emotional Skills (SES) (mean ± SD)	42.5 ± 7.7
Quality of Life (mean ± SD)	69.2 ± 12.4

**Table 2 ijerph-17-07796-t002:** Pearson’s correlation coefficients of variables.

	1	2	3	4	5	6	7	8
1. Quality of Life	…							
2. SES	0.45 ***	…						
3. Executive Function	0.64 ***	0.52 ***	…					
4. Mindfulness	0.39 **	0.14	0.54 ***	…				
5. Age	0.08	−0.12	0.03	0.01	…			
6. Gender (Male)	0.08	−0.09	0.08	0.13	−0.08	…		
7. First-Generation	0.09	0.00	0.04	0.04	0.33 *	−0.01	…	
8. Second-Generation	−0.22	0.02	−0.06	−0.09	−0.28	0.05	−0.79 ***	…

Note: N = 96. * *p* < 0.05, ** *p* < 0.01, *** *p* < 0.001.

**Table 3 ijerph-17-07796-t003:** Decomposition of Standardized Effects in SEM.

Predictor	Dependent Variable	Direct Effect	Indirect Effect	Total Effect
Mindfulness	Executive Function	0.53 ***	---	0.53 ***
	SES	---	0.29 ***	0.29 ***
	Quality of Life	---	0.33 ***	0.33***
Executive Function	SES	0.54 ***	---	0.54 ***
	Quality of Life	0.51 ***	0.11 *	0.62 ***
SES	Quality of Life	0.20 ***	----	0.20 *
Age	Executive Function	0.02	---	0.02
	SES	−0.15	0.01	−0.14
	Quality of Life	0.08	−0.02	0.06
Male	Executive Function	0.02	----	0.02
	SES	−0.15	0.01	−0.14
	Quality of Life	0.08	−0.02	0.06
First Generation	Executive Function	0.01	---	0.01
	SES	0.11	0.01	0.12
	Quality of Life	−0.25 *	0.03	−0.22
Second Generation	Executive Function	0.00	---	0.00
	SES	0.11	−0.00	0.11
	Quality of Life	−0.37 **	0.02	−0.35 *

Note: N = 96. * *p* < 0.05, ** *p* < 0.01, *** *p* < 0.001.

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
