# Peer review of "Associations between Mindfulness, Executive Function, Social-Emotional Skills, and Quality of Life among Hispanic Children"

_ijerph, 2020, doi:10.3390/ijerph17217796_

Round 1

Reviewer 1 Report

The paper is scientifically accurate and it fits well the scope of the IJERPH. The results are important for both scientific community and the public. The topic is novel and interesting. The title is informative. Abstract is informative and precise. Results are summarized sufficiently in both – abstract and main text. The language of the paper is scientific but quite simple, clear and understandable. The number of analysed children is quite high. The statistics is good. The number of figures and tables is appropriate.

Minor comments - to consider:

  • include reference: Łoś et al: Relationship between executive functions, mindfulness, stress and performance (IJERPH, 2020) in the introduction or discussion (?)
  • Figures 1 and 2 are not clear, should be improved before or during publishing service (including arrows !)
  • In Table 1 - column two is not clear for the reader - percentages or mean - there should be a mark "%" included after the percentages; I suggest to rebuild this Table for better readability

I recommend this paper for rapid publication.

Reviewer 2 Report

please refer to the attached file

Reviewer 3 Report

This is an interesting paper that could have an impact on educational policy and practice, down the line. However, one of the major problems is the title which is misleading - given the design of the study you are looking at relationships/correlations and not at causality, as this is a cross-sectional/observation study and not an experimental/longitudinal one. You mention this at the end, but still keep the causal language throughout. 

I think this paper should be published, after the following have been taken into account:

The abstract needs restructuring to include a bit of information on the methods – what did you use, what did you do ,what data were collected?

The introduction and literature review is well written and logically builds to the presentation of the hypothesis. Given the quantitative nature of the investigation, it would be better to articulate the hypothesis as a statement (or even as a null hypothesis) that can be supported or refuted.

Lines 96 to 99 are not clear. What is the ‘full sample’? Did you check for differences in the Hispanic and non-Hispanic sample? It sounds like you did and found no difference, but it is not clear. Please clarify.

In future consider undertaking a power analysis to estimate the minimum sample size prior to recruitment.

It would be good to know some of the internal validity measures for the scales used (e.g. Cronbach’s alpha), reported in the validating studies, so that they can be compared to that of your own study.

Line 190-191 – is there a reason why younger pupils are more likely to be first generation? If not, could it be a chance finding, especially given the small sample size?

It is worth looking at a multiple comparison correction, given how many relationships you are looking at in your inferential statistics. Looking at your values, most of your correlations will still be significant even after a conservative multiple testing correction, like Bonferroni.

Some of the literature introduced in the discussion (lines 280 onwards) would have been useful when looking at mindfulness in the literature review part of the paper, where you gave a very brief definition and no in-depth explanation of what you perceive the mechanism of action to be in relation to executive function etc. You explain it very well in the second part of the paper, but a little more depth at the start would be useful.

301-302 – as stated above, you are talking about an effect, and therefore causality, as you do in the title. Can you ascertain that from an observational study such as this one, rather than an experimental or longitudinal one? You mention these limitations at the end, yet claim to still be looking at an effect. The paper would be better placed titled ‘the relationship between’ rather than ‘the effect of’. Please consider using correct language both in the title and in the text of this paper.

Round 2

Reviewer 2 Report

further suggestions:

the authors have addressed majority of my concerns raised in the first review. there is one more suggestion regarding the conclusion:

according to the revised version, conclusion includes three long paragraphs which summarie and explain the findings, and present the limitations of the study. As conclusion is used to be a short paragraph summarise the major findings and implication, it is recommended to put the last two paragraphs (lines 361-381) to the font and leave line352-360 as conclusion.

Author Response

Thank you again for your constructive and helpful comments. Accordingly, we have moved the last two paragraphs (lines 361-381) to the end of discussion and only use line 352-360 as conclusion. We appreciate your valuable help on the revision.